# SEAHORSE: A Multilingual, Multifaceted Dataset for Summarization Evaluation

**Elizabeth Clark**[1]  **Shruti Rijhwani**[1]  **Sebastian Gehrmann**[2]  **Joshua Maynez**[1]
**Roee Aharoni**[2]  **Vitaly Nikolaev**[1]  **Thibault Sellam**[1]  **Aditya Siddhant**[1]
**Dipanjan Das**[1]  **Ankur P. Parikh**[1]
[1]Google DeepMind  [2]Google Research
Contact: eaclark@google.com

## Abstract

Reliable automatic evaluation of summarization systems is challenging due to the multifaceted and subjective nature of the task. This is especially the case for languages other than English, where human evaluations are scarce. In this work, we introduce SEAHORSE, a dataset for multilingual, multifaceted summarization evaluation. SEAHORSE consists of 96K summaries with human ratings along 6 dimensions of text quality: comprehensibility, repetition, grammar, attribution, main ideas, and conciseness. SEAHORSE covers 6 languages, 9 systems (including the reference text), and 4 summarization datasets. As a result of its size and scope, SEAHORSE can serve both as a benchmark to evaluate learnt metrics, as well as a large-scale resource for training such metrics. We show that metrics trained with SEAHORSE achieve strong performance on two out-of-domain meta-evaluation benchmarks: TRUE (Honovich et al., 2022) and mFACE (Aharoni et al., 2023). We make the SEAHORSE dataset and metrics publicly available for future research on multilingual and multifaceted summarization evaluation.[1]

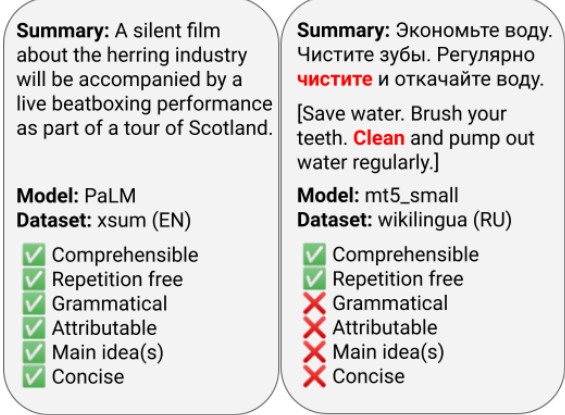

Figure 1: Two summaries from the SEAHORSE dataset paired with human ratings for 6 dimensions of quality. In the second summary, the word in **bold** has a grammatical error in Russian; it uses the wrong aspect. The rater has noted this error, along with several others.

## 1 Introduction

Evaluating the quality of generated text is an increasingly difficult problem as large language models produce text of rapidly improving quality (Radford et al., 2019; Ouyang et al., 2022; Chowdhery et al., 2022). In spite of the improvements, such models often generate text that includes hallucinations and other subtle errors (Wiseman et al., 2017; Maynez et al., 2020; Parikh et al., 2020; Ji et al., 2023; Borji, 2023), making reliable evaluation essential for driving progress.

Common n-gram metrics such as BLEU (Papineni et al., 2002) and ROUGE (Lin, 2004) are often not well correlated with human judgments

for many natural language generation (NLG) tasks such as machine translation (Kocmi et al., 2021; Freitag et al., 2021a), summarization (Kryscinski et al., 2020), and dialogue (Dziri et al., 2022). Consequently, human evaluation is often necessary to reliably evaluate NLG systems. However, designing human annotation pipelines and obtaining annotations is resource-intensive, time-consuming, and not easily reproducible. Developing more reliable automatic evaluation metrics would make model development faster and more efficient. With this in mind, much recent work has focused on learnt metrics, i.e., neural classification or regression models that aim to directly predict scores that evaluate the quality of generated text (Zhang* et al., 2020; Sellam et al., 2020; Rei et al., 2020; Liu et al., 2023), often trained with human ratings.

As a result, large-scale collections of human evaluations serve two critical roles in NLG metric development: (1) a source of training data for learnt metrics and (2) a meta-evaluation benchmark for the performance of these learnt metrics. The

---

[1]Data and metrics are available at https://goo.gle/seahorse

large potential of such datasets is exemplified by the WMT metrics shared task,[2] which has enabled rapid development of learnt metrics for machine translation that exhibit considerably higher correlation to human judgment than BLEU (Bojar et al., 2016; Freitag et al., 2021b).

However, outside of machine translation, the existence of such collections of human judgments is limited. Human annotations collected in NLG evaluations are rarely released (Gehrmann et al., 2022), and even when they are, they tend to cover a single language (typically English) and are from a single dataset or task, limiting the robustness of models and metrics trained on these annotations. Moreover, such annotations are often based on the test split of existing datasets (e.g., Fabbri et al., 2021; Aharoni et al., 2023), which can be problematic for training learnt metrics. This is because the primary advantage of reliable automatic evaluation is to help model development, e.g., hyperparameter selection on the validation set; therefore a neural metric trained on test set annotations would, in general, lead to overfitting.

In this work, we propose SEAHORSE,[3] a large-scale dataset for multilingual summarization evaluation. Our dataset consists of 96K summaries with ratings along 6 quality dimensions: comprehensibility, repetition, grammar, attribution, main ideas, and conciseness, in 6 languages, for 9 systems (8 models plus the human-authored reference summaries) across 4 summarization datasets (see examples in Figure 1). The training and validation splits of the dataset come from the validation sets of the original summarization corpora to prevent test set contamination when training metrics. This permits us to train a learnt metric for each quality dimension that can be used for offline model evaluation.

We evaluate the metrics learned from SEAHORSE on the SEAHORSE test set, as well as other existing meta-evaluation benchmarks, such as mFACE (Aharoni et al., 2023) and TRUE (Honovich et al., 2022). Our experiments show that the metrics generalize across datasets, tasks, and languages. For example, we demonstrate that although SEAHORSE includes data in 6 languages, the resulting learnt metrics achieve strong performance on the mFACE benchmark, which consists of 45 languages, exhibiting their zero-shot multi-lingual generalization potential. To summarize, the contributions of this paper are:

- We conduct a comprehensive, large-scale human evaluation for summarization across six languages, six quality facets, nine systems and four datasets, resulting in over 96K human-rated summaries. To the best of our knowledge, this is the largest multilingual, multi-faceted summarization evaluation resource.

- We train a learnt metric for each of the evaluated quality facets, and show that the metrics outperform strong baselines across our in-domain test set and previously published out-of-domain benchmarks, highlighting the quality of the human annotations we collect and the broad utility of our learnt metrics.

- We release our dataset and metrics to foster future work on multilingual, multifaceted summarization.

## 2 The SEAHORSE dataset

The SEAHORSE dataset consists of 96,645 summaries annotated with human ratings along 6 quality dimensions. In this section, we describe the SEAHORSE dataset, how we generated the summaries, and how we collected the annotations.

### 2.1 The summaries

The examples in SEAHORSE are in 6 languages: German (de), English (en), Spanish (es), Russian (ru), Turkish (tr), and Vietnamese (vi). We chose these languages by considering geographic and typological diversity and the availability of summarization datasets in those languages.

The summaries are based on articles from 4 different datasets in the GEM benchmark (Gehrmann et al., 2021):

- **XSum** (Narayan et al., 2018): An English dataset where the task is to generate a one-sentence summary of a BBC News article.

- **XL-Sum** (Hasan et al., 2021): Similar to XSum, the goal of this dataset is to generate a single-sentence summary of a BBC news article, but it covers 44 languages excluding English.

- **MLSum** (Scialom et al., 2020): A summarization dataset obtained from online newspapers in 5 languages.

---

[2] https://wmt-metrics-task.github.io/

[3] SEAHORSE stands for *SummariEs Annotated with Human Ratings in Six languagEs*.

| language | dataset | articles | annotations |
|---|---|---|---|
| de | mlsum | 3359 | 7506 |
| | wikilingua | 2999 | 7085 |
| en | xsum | 894 | 6651 |
| | xlsum | 2433 | 7884 |
| | wikilingua | 2383 | 7804 |
| es | xlsum | 2231 | 4890 |
| | mlsum | 2235 | 4857 |
| | wikilingua | 2183 | 5002 |
| ru | xlsum | 3298 | 7254 |
| | wikilingua | 2948 | 7288 |
| tr | xlsum | 2186 | 10627 |
| | wikilingua | 770 | 4791 |
| vi | xlsum | 2497 | 7522 |
| | wikilingua | 1951 | 7484 |

Table 1: The number of unique articles and the number of annotated summaries collected from each dataset to create SEAHORSE. Each article is summarized by several different summarization systems, which were evaluated by human annotators.

- **WikiLingua** (Ladhak et al., 2020): A dataset in 18 languages where the goal is to summarize how-to guides from WikiHow.

A breakdown of SEAHORSE across languages and datasets is in Table 1.

For each dataset, we randomly selected articles from their validation splits to comprise the SEAHORSE training and validation sets, and articles from the test splits to make up the SEAHORSE test set. This distinction is important when using the dataset for training evaluation metrics (discussed in §4), because learnt metrics are typically used for model development, and hyperparameter selection is done on the validation set. Using a metric that was trained on test data would lead to overfitting. Our dataset construction ensures that a learnt metric can be trained on SEAHORSE data without concerns of test set leakage.

Next, we generate summaries for each article in the dataset. The summaries come from a subset of 9 different systems, which we will denote as follows:

- **reference**: The human-authored summaries associated with each article from the original datasets.

- **t5_base**: The 220M-parameter version of the T5 model (Raffel et al., 2020). (This model is English-only, so we only use it to generate summaries with our en datasets.)

- **t5_base_250**: The t5_base model with an under-trained checkpoint, trained for only 250 steps (en only).

- **t5_xxl**: The 11B-parameter version of T5 (en only).

- **mt5_small**: The 300M-parameter version of mT5 (Xue et al., 2021).

- **mt5_small_250**: The same mt5_small model but using the checkpoint after training 250 steps.

- **mt5_xxl**: The 13B-parameter mT5 model.

- **palm_1shot**: 540B-parameter PaLM model (Chowdhery et al., 2022) prompted with one in-domain example.

- **palm_finetuned**: 540B-parameter PaLM model (Chowdhery et al., 2022) finetuned on training data for the respective dataset.

Our choice of systems covers a range of expected system performances in order to capture a large diversity of system outputs and model error types. For instance, an under-trained small model (**mt5_small_250**) would likely have different errors than a 1-shot large language model (**palm_1shot**). Details about how the summaries are generated from these models are in Appendix A.

### 2.2 Annotation methodology

For each summary, we collect annotations along 6 dimensions, also referred to as Q1–6:

**Q1 comprehensible**: The summary can be read and understood by the rater. (If "No," the rest of the questions will be skipped.)

**Q2 repetition**: The summary is free of unnecessarily repeated information.

**Q3 grammar**: The summary is grammatically correct.

**Q4 attribution**: All the information in the summary is fully attributable to the source article, as defined in Rashkin et al. (2021).

**Q5 main ideas**: The summary captures the main idea(s) of the source article.

**Q6 conciseness**: The summary concisely represents the information in the source article.

For the first 3 questions, annotators see only the summary. The article is revealed when the raters are answering questions 4–6. They can answer "Yes," "No," or "Unsure" to each question and have the option to leave comments or flag any issues they see in the article. The annotation interface is shown in Figure 2.

Note that our annotation process is *reference-less*, i.e., the annotator is never comparing a model-generated summary with the reference summary. They evaluate each summary on its own. Given the subjectivity of summarization, we believe this approach allows us to adequately reward models that generate relevant summaries that may be different than the reference. Moreover, this enables us to train reference-less metrics in §4, which have an added benefit of being able to be used at inference time for re-ranking.

The raters are paid, full-time annotators who were trained for this specific task and worked under the supervision of a project manager. For the non-English languages, the raters are bilingual, proficient in both the annotation language and English. They received a detailed set of instructions in English describing the 6 dimensions of quality and positive and negative examples of each in the target language. We created a set of 109 summaries with gold ratings, which we used to train the raters. Each annotator rated 20–30 summaries from this gold set. If the rater performed well on this subset, they were qualified to move forward with the annotation task. Otherwise, the annotator received feedback and were asked to complete another 10–20 ratings. This training process was repeated as needed.

A small number of approved annotators were removed during the annotation process, due to issues flagged by the annotation team and the authors. The ratings from the removed annotators are not included in the dataset.

## 3   Dataset analysis

We first analyze the dataset's composition and the quality of the collected annotations. Table 2 contains the median length of summaries produced by each model, along with two measures of the overlap between the summaries and the source articles.

| model | length | rouge | 20% copy |
|---|---|---|---|
| reference | 227 | 20.26 | 0.00 |
| t5_base_250 | 92 | 20.95 | 0.00 |
| t5_base | 101 | 22.02 | 0.02 |
| t5_xxl | 115 | 21.65 | 0.01 |
| mt5_small_250 | 128 | 21.33 | 0.02 |
| mt5_small | 171 | 21.81 | 0.04 |
| mt5_xxl | 194 | 20.77 | 0.01 |
| palm_1shot | 254 | 27.34 | 0.14 |
| palm_finetuned | 194 | 20.97 | 0.01 |

Table 2: The median number of characters (`length`), ROUGE-L between the summary and article (`rouge`), and the proportion of summaries where the first 20% of the summary exactly matches the beginning of the source article (`20% copy`) for all the summaries generated by each model.

| Model | Q1 | Q2 | Q3 | Q4 | Q5 | Q6 |
|---|---|---|---|---|---|---|
| reference | 0.97 | 0.97 | 0.91 | 0.54 | 0.68 | 0.43 |
| t5_base_250 | 0.97 | 0.79 | 0.91 | 0.41 | 0.42 | 0.25 |
| t5_base | 0.98 | 0.92 | 0.93 | 0.59 | 0.59 | 0.43 |
| t5_xxl | 0.99 | 0.97 | 0.95 | 0.65 | 0.67 | 0.51 |
| mt5_small_250 | 0.71 | 0.43 | 0.59 | 0.27 | 0.19 | 0.1 |
| mt5_small | 0.86 | 0.57 | 0.73 | 0.36 | 0.35 | 0.19 |
| mt5_xxl | 0.96 | 0.94 | 0.88 | 0.55 | 0.65 | 0.43 |
| palm_1shot | 0.88 | 0.85 | 0.79 | 0.71 | 0.57 | 0.47 |
| palm_finetuned | 0.98 | 0.98 | 0.9 | 0.69 | 0.71 | 0.56 |

Table 3: The percent of "Yes" responses, broken down by model and question.

The 1-shot PaLM model is particularly likely to copy from the article as its output, obtaining the highest ROUGE-L[4] (Lin, 2004) scores between the summary and the article. In 14% of cases, the beginning of the 1-shot summaries (the first 20% of the summary) exactly matched the beginning of the reference article.

Table 3 shows the percent of summaries from each summarization system that received a positive (i.e., "Yes") rating from annotators. While there is variation across models and datasets, most summaries are rated positively for questions 1–3 (comprehensibility, repetition, and grammar). The rate of positive responses drops for questions 4–6 (attribution, main ideas, and conciseness), indicating that these areas remain a challenge for summarization models. A more detailed break down of the positive response rates is in Appendix B.

Note that the reference summaries do not always receive the highest rate of positive responses. The

---

[4]All ROUGE scores in this paper are calculated with SentencePiece tokens: https://github.com/google/sentencepiece

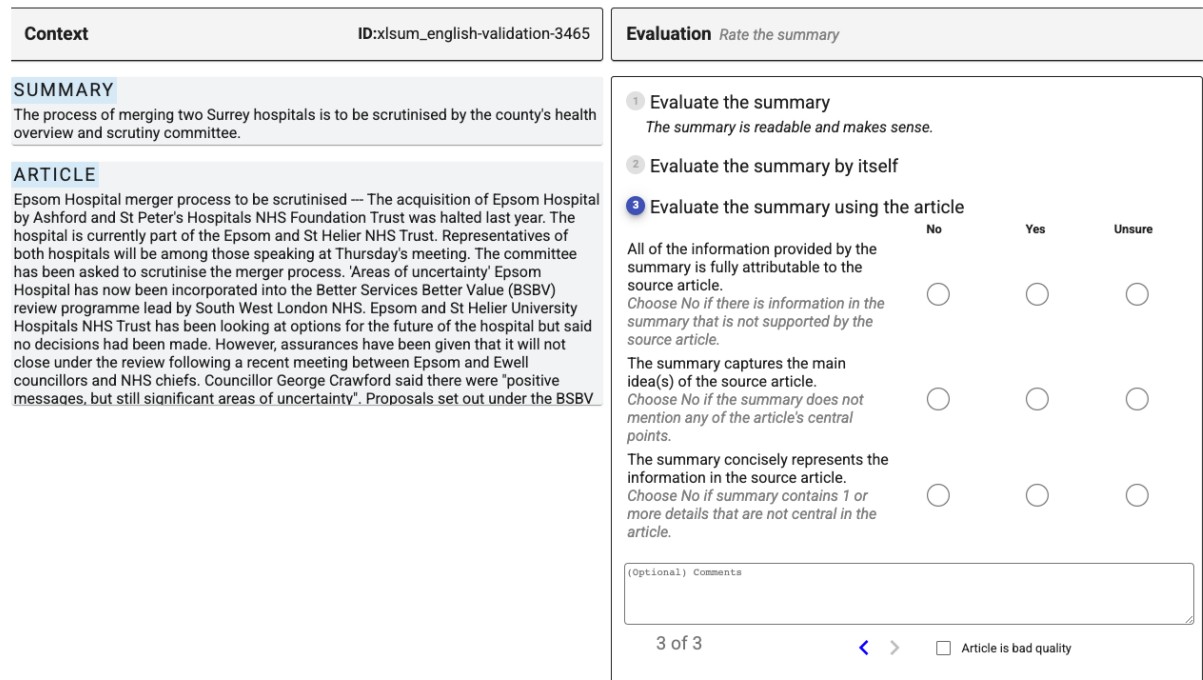

Figure 2: The annotation interface used to collect SEAHORSE. First, Question 1 and the summary are shown to the evaluator. Once they confirm that the summary is comprehensible, Questions 2–3 are shown. Finally, the article and Questions 4–6 are displayed (as pictured above).

way in which reference texts are collected may limit their quality along some dimensions. For example, the text that was collected as a reference summary may not have been intended to be read as a standalone replacement for the article, and therefore may not be fully attributable to the article, as Rashkin et al. (2021) point out.

We can use the positive response rates to inspect the quality of the dataset by verifying the presence of 3 patterns we expect to see in the data: 1) higher positive response rates for better summarization models, 2) high correlation between the responses to Q4&6 and Q5&6, and 3) annotator agreement across the 6 dimensions.

**Order of model quality**  Our first expectation is that summaries generated by better summarization models should receive more positive responses from raters. We have 3 types of model pairs where we can expect one model to generate better summaries than the other: 1) a larger model should outperform a smaller model (the xxl vs. the small model), 2) a fully trained model should outperform an under-trained model (the small vs. the small_250 model), and 3) a finetuned model should outperform a 1-shot prompted model (the finetuned vs. 1-shot PaLM models).

We compare how often these model pairs pro-duce low-quality summaries, i.e., summaries that are unintelligible to readers. In Table 3, we see that mt5_xxl produces fewer incomprehensible (Q1) summaries than mt5_small, which produces fewer than mt5_small_250. The same holds true for the T5 models, and palm_finetuned produces fewer incomprehensible summaries than palm_1shot, reflecting the expected relationship in quality between model pairs. While these results are averaged over the entire dataset, we see the same result when controlling for the source article and only considering items that have summaries generated by all 9 systems (see Appendix B).

This pattern generally holds across the other dimensions of quality as well. There is one notable exception: PaLM's perfomance on attribution (Q4). For 4 languages, palm_1shot is more often rated as being faithful to the input article than palm_finetuned, which is likely due to its tendency to copy the article directly.

Generally, however, the SEAHORSE ratings capture the relative differences in model quality we expect to see when evaluating two models with known differences.

**Correlation between dimensions**  Conciseness (Q6) is related to two other dimensions in our annotation: attribution (Q4) and main ideas (Q5). A

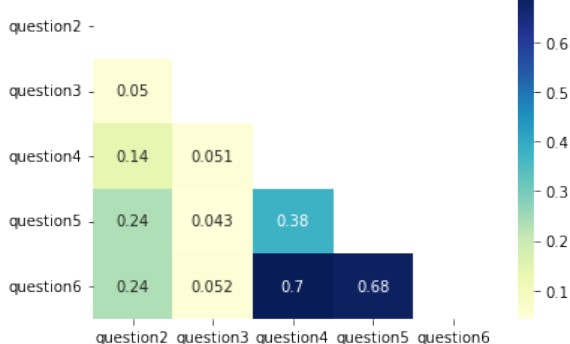

Figure 3: The Pearson correlation between responses for questions 2-6.

| Lang | Avg | Q1 | Q2 | Q3 | Q4 | Q5 | Q6 |
|------|------|------|------|------|------|------|------|
| de | 0.84 | 0.97 | 0.98 | 0.95 | 0.81 | 0.67 | 0.66 |
| es | 0.82 | 0.92 | 0.97 | 0.83 | 0.74 | 0.7 | 0.74 |
| en | 0.81 | 0.97 | 0.94 | 0.95 | 0.69 | 0.61 | 0.69 |
| ru | 0.82 | 0.86 | 0.97 | 0.88 | 0.71 | 0.73 | 0.76 |
| tr | 0.82 | 0.93 | 0.96 | 0.86 | 0.74 | 0.7 | 0.74 |
| vi | 0.81 | 0.95 | 0.98 | 0.88 | 0.68 | 0.66 | 0.69 |
| avg | 0.82 | 0.93 | 0.97 | 0.89 | 0.73 | 0.68 | 0.72 |

Table 4: The average pairwise agreement, broken down by language and question.

summary cannot be considered a "concise representation of the information in the article" if it has information that is not in the article (i.e., a "No" response for Q4) or if does not represent the main points in the article (i.e., a "No" response for Q5), which was a detail pointed out to evaluators in the task instructions. Therefore, we expect Q6 to be positively correlated with both of these dimensions if the annotators understood the task and the relationship between the dimensions of quality.

In $> 99\%$ of cases when the annotator says a summary is not attributable (Q4) or they say it lacks the main ideas from the article (Q5), they also say it is not concise (Q6). This is also reflected in Figure 3, which shows that the strongest correlation between questions is between questions 4&6 and questions 5&6. These results show the pattern we expect to see in the data given the task definition and instructions, and it demonstrates the annotators' ability to understand and execute the annotation task.

| Q1 | Q2 | Q3 | Q4 | Q5 | Q6 |
|------|------|------|------|------|------|
| 0.49 | 0.87 | 0.35 | 0.47 | 0.4 | 0.41 |

Table 5: Krippendorff's $\alpha$ by question.

**Annotator agreement** While most items in the dataset were annotated once, we collected 2 additional ratings for a subset of the data to compare annotators' scores. Out of 8,920 duplicated annotations, the overall pairwise agreement between raters was 82%. Table 4 breaks down the pairwise accuracy across all languages and questions. Questions 1–3 have higher agreement, while questions 4–6 (which depend on more context and have a higher degree of subjectivity) have lower agreement. A similar trend is reflected in the Krippendorff's $\alpha$ values (Krippendorff, 1980, shown in Table 5), which correct for the probability of random agreement, except grammar (Q3) scores lowest.

These patterns in the annotators' responses are positive indicators about the overall quality of the SEAHORSE ratings. However, the more important test of the dataset's quality is its usefulness for developing evaluation metrics, which we discuss in the next section.

## 4 Learning and evaluating metrics with SEAHORSE

The SEAHORSE dataset is meant to serve both as a source of training data for learnt metrics as well as a meta-evaluation benchmark for these metrics. In this section, we evaluate SEAHORSE on these aspects by looking at how well metrics finetuned with our collected annotations can predict human ratings of generated summaries, both from the SEAHORSE test set and other existing datasets. When training metrics, we use a filtered version of the dataset that removes all duplicates and non-Yes or No ratings (88,280 total items). We divide the annotations into train/dev/test splits, where the summaries in the train and dev sets are based on articles from the original datasets' validation sets. The test set of SEAHORSE contains summaries of the articles in the original datasets' test sets.

### 4.1 Metrics

One way to train a metric using SEAHORSE is to finetune a text-to-text generation model, where the model is trained to take an article and summary as its input and to output the string '0' or '1' as a prediction of the human rating. We finetune mT5-_xxl (Xue et al., 2021) with the SEAHORSE training set to do this task, finetuning a separate metric for each dimension of quality. We call this model

$\text{mt5}_{\text{SEAHORSE}}$[5]. More details are in Appendix A. Note that our goal is not to train a state-of-the-art metric but rather to evaluate the utility of SEA-HORSE as a resource to train and evaluate such metrics.

We compare the performance of $\text{mt5}_{\text{SEAHORSE}}$ to several baselines:

- **majority_class** A majority class baseline (i.e., picking the most frequent class).

- **ROUGE-L** The ROUGE-L score between the article and the summary.

Specifically for the attribution (Q4) task, we consider a third baseline approach; attribution is closely related to natural language inference (NLI) (Fyodorov et al., 2000; Dagan et al., 2006), and Honovich et al. (2022) show that models fine-tuned on NLI data perform well as faithfulness metrics. Therefore we consider two variants of an NLI-based baseline:

- **t5**$_{\text{NLI}}$: An English NLI model proposed by Honovich et al. (2022).[6] T5_xxl is fine-tuned on the following datasets: SNLI (Bowman et al., 2015), MNLI (Williams et al., 2018), Fever (Thorne et al., 2018), Scitail (Khot et al., 2018), PAWS (Zhang et al., 2019), and VitaminC (Schuster et al., 2021).

- **mt5**$_{\text{XNLI}}$: A multilingual version, where mT5_xxl is finetuned on XNLI (Conneau et al., 2018).

We note that since we are operating in the reference-free setting, other learnt metrics such as BLEURT (Sellam et al., 2020) or BERTScore (Zhang* et al., 2020) are not applicable since they measure the similarity between the prediction and reference.

We evaluate the SEAHORSE and baseline metrics in two ways: the area under the ROC curve and the correlation (Pearson's $\rho$) between the metric and human scores. These measures are not sensitive to a thresholding value and are also used in the work we compare with (Honovich et al., 2022; Aharoni et al., 2023).

### 4.2 Evaluation on the SEAHORSE test set

We first evaluate $\text{mt5}_{\text{SEAHORSE}}$ on the SEAHORSE test set to confirm that a model is able to learn

to predict the different dimensions of quality in SEAHORSE. The results are shown in Table 6. As expected, we see that the $\text{mt5}_{\text{SEAHORSE}}$ model is able to predict SEAHORSE ratings better than the baselines according to both our metrics. The repetition (Q2) metric performs the best out of the 6 dimensions, which is also the dimension with the highest pairwise annotator agreement. Examples of summaries paired with human, SEAHORSE, and ROUGE-L ratings can be found in Appendix C.

Reducing the size of the base mT5 model from XXL (13B parameters) to Large (1.2B) drops the performance of the metric, but shows similar trends and still outperforms all baseline approaches. More $\text{mt5\_L}_{\text{SEAHORSE}}$ results can be found in Appendix D.

### 4.3 Evaluation on the mFACE dataset

In addition to achieving good performance on the SEAHORSE test set, we would like to evaluate how well models trained on SEAHORSE generalize to other multilingual summarization human evaluation datasets without any further tuning. This would give evidence that improving on SEAHORSE would lead to better evaluation metrics in general.

For this purpose, we choose the mFACE dataset[7] (Aharoni et al., 2023). mFACE contains human evaluations of the XL-Sum test set, which consists of 45 languages on 3 dimensions: quality, attribution, and informativeness. While their definition of attribution is the same as ours (i.e., following AIS (Rashkin et al., 2021)), their definitions of quality (*Is the summary comprehensible?*) and informativeness (*Is the summary a good summary of the article?*) do not line up exactly with a single one of our questions, a misalignment that we expect to occur in practice given the lack of standardization of summarization human evaluation.

As a result, for each mFACE dimension, we use the SEAHORSE metric for the question that is most similar; attribution clearly aligns with Q4, and for quality and informativeness, we consider Q1 and Q6 to be the closest fit, respectively.

We evaluate on both the full mFACE dataset (all languages), as well as the 5-language subset that is common to both mFACE and SEAHORSE (en, es, ru, tr, vi). In addition to our baseline models, we also compare to an "upper-bound" mT5_xxl model that has been directly trained on mFACE data ($\text{mt5}_{\text{MFACE}}$).

---

[5]There are actually 6 different models, one for each question, but we use the notation $\text{mt5}_{\text{SEAHORSE}}$ for simplicity.

[6]https://huggingface.co/google/t5_xxl_true_nli_mixture

---

[7]We obtained the dataset by contacting the authors.

| Metric | Q1 ρ | Q1 roc | Q2 ρ | Q2 roc | Q3 ρ | Q3 roc | Q4 ρ | Q4 roc | Q5 ρ | Q5 roc | Q6 ρ | Q6 roc |
|---|---|---|---|---|---|---|---|---|---|---|---|---|
| majority_class | - | 0.5 | - | 0.5 | - | 0.5 | - | 0.5 | - | 0.5 | - | 0.5 |
| ROUGE-L | 0.04 | 0.54 | 0.06 | 0.54 | -0.03 | 0.43 | 0.13 | 0.55 | 0.03 | 0.53 | 0.02 | 0.54 |
| mt5$_{XNLI}$ | - | - | - | - | - | - | 0.43 | 0.78 | - | - | - | - |
| mt5_L$_{SEAHORSE}$ | 0.44 | 0.88 | 0.74 | 0.97 | 0.37 | 0.81 | 0.55 | 0.82 | 0.46 | 0.78 | 0.45 | 0.77 |
| mt5$_{SEAHORSE}$ | **0.52** | **0.90** | **0.86** | **0.98** | **0.45** | **0.84** | **0.59** | **0.85** | **0.50** | **0.80** | **0.52** | **0.81** |

Table 6: Metrics' ability to predict SEAHORSE ratings, measured with Pearson's coefficient ($\rho$) and the area under the ROC curve (roc). mt5_L$_{SEAHORSE}$ is a finetuned version of mT5_large; the other mt5 metrics finetune mT5_xxl.

| | | mFACE - 5 languages | | | | | | mFACE - all languages | | | | | |
| | | Quality | | Attribution | | Informativeness | | Quality | | Attribution | | Informativeness | |
| | Metric | $\rho$ | roc | $\rho$ | roc | $\rho$ | roc | $\rho$ | roc | $\rho$ | roc | $\rho$ | roc |
|---|---|---|---|---|---|---|---|---|---|---|---|---|---|
| *Not trained on mFACE* | majority_class | - | 0.5 | - | 0.5 | - | 0.5 | - | 0.5 | - | 0.5 | - | 0.5 |
| | ROUGE-L | 0.02 | 0.51 | 0.14 | 0.58 | 0.06 | 0.56 | 0.06 | 0.52 | 0.09 | 0.52 | 0.09 | 0.52 |
| | mt5$_{XNLI}$ | - | - | 0.45 | **0.82** | - | - | - | - | 0.34 | 0.74 | - | - |
| | mt5$_{SEAHORSE}$ | **0.09** | **0.73** | **0.50** | 0.79 | **0.50** | **0.81** | **0.15** | **0.70** | 0.52 | 0.81 | 0.40 | **0.74** |
| *Trained on mFACE* | mt5$_{MFACE}$ | **0.25*** | 0.68 | **0.51*** | 0.81 | 0.47 | 0.79 | **0.35*** | 0.61 | **0.52*** | **0.82*** | **0.47*** | **0.80*** |

Table 7: Metrics' ability to predict mFACE ratings, measured with Pearson's coefficient ($\rho$) and the area under the ROC curve (roc). The asterisk indicates that the associated model was trained on the training portion of the mFACE dataset.

Results are shown in Table 7. In all but one column, mt5$_{SEAHORSE}$ outperforms the other methods that were not trained on the mFACE data and also performs well on the languages it was not finetuned on. mt5$_{SEAHORSE}$ even performs comparably to mt5$_{MFACE}$ on the 5 language subset on all dimensions, and the attribution dimension on the all-language set. mt5$_{MFACE}$ performs better on quality and informativeness on the all-language set, as one would expect, since it has seen supervised data from those languages and dimensions whereas mt5$_{SEAHORSE}$ is applied in a zero-shot setting.

### 4.4 Evaluation on the TRUE Benchmark

Finally, we focus on the attribution dimension of quality, since issues of faithfulness in generated text are increasingly important (Wiseman et al., 2017; Tian et al., 2019; Zhou et al., 2021; Dziri et al., 2022; Ji et al., 2023). The TRUE benchmark (Honovich et al., 2022) consists of several English datasets across summarization, dialogue, verification, and paraphrasing: FRANK (Pagnoni et al., 2021), SummEval (Fabbri et al., 2021), MNBM (Maynez et al., 2020), QAGS (Wang et al., 2020), BEGIN (Dziri et al., 2022), $Q^2$ (Honovich et al., 2021), DialFact (Gupta et al., 2022), FEVER (Thorne et al., 2018), VitaminC (Schuster et al., 2021), and PAWS (Zhang et al., 2019).

As in the prior section, we apply mt5$_{SEAHORSE}$ without any further finetuning to these datasets to assess its ability to evaluate attribution to other datasets and tasks beyond summarization. In addition to comparing to the majority class and ROUGE-L baselines, we also compare with t5$_{NLI}$.

Results are shown in Table 8. mt5$_{SEAHORSE}$ achieves the best results across the summarization datasets, which is expected as many of these datasets consist of XSum and CNN/DailyMail (Hermann et al., 2015), the first of which is also a source of the SEAHORSE summaries and the second is a different news summarization dataset. Interestingly, despite only being trained on summarization data, mt5$_{SEAHORSE}$ performs competitively to t5$_{NLI}$ on the dialogue datasets (BEGIN, $Q^2$, and DialFact), indicating its suitability for evaluating tasks outside of summarization. t5$_{NLI}$ performs best on the Fever, VitaminC, and PAWS tasks, which is expected given that the t5$_{NLI}$ model was trained on these datasets.

## 5 Related work

We briefly review other large-scale datasets of human evaluations of summaries that have been released and compare them to SEAHORSE, but note that most focus on annotating the test data, which would lead to test data contamination when training metrics.

SummEval (Fabbri et al., 2021) and RealSumm (Bhandari et al., 2020) are summarization meta-evaluation benchmarks with 12,800 and 7,742 annotations respectively. These benchmarks focus on a single language and single dataset: the

| | FRANK | SummEval | MNBN | QAGS-C | QAGS-X | BEGIN | $Q^2$ | DialFact | Fever | VitaminC | PAWS |
|---|---|---|---|---|---|---|---|---|---|---|---|
| majority_class | 0.5 | 0.5 | 0.5 | 0.5 | 0.5 | 0.5 | 0.5 | 0.5 | 0.5 | 0.5 | 0.5 |
| ROUGE-L | 0.55 | 0.57 | 0.53 | 0.44 | 0.55 | 0.63 | 0.54 | 0.49 | 0.48 | 0.50 | 0.60 |
| mT5$_{\text{SEAHORSE}}$ | **0.94** | **0.87** | **0.83** | **0.91** | **0.87** | 0.84 | 0.82 | 0.87 | **0.91** | **0.78** | **0.82** |
| T5$_{\text{NLI}}$ | 0.90 | 0.79 | 0.76 | 0.77 | 0.85 | **0.85** | **0.83** | 0.92 | **0.95\*** | **0.98\*** | **0.99\*** |

Table 8: Metrics' performance on the TRUE benchmark, measured with area under the ROC curve. t5$_{\text{NLI}}$ is a T5-xxl model trained on a mixture of NLI datasets that includes the FEVER, VitaminC, and PAWS training sets (and thus those numbers are indicated with an asterisk).

CNN/DailyMail English summarization dataset. The RoSE benchmark (Liu et al., 2022) contains 22K summary-level annotations across 3 summarization datasets, including a subset from the CNN/DailyMail validation set, and Stiennon et al. (2020) released 65K summary comparisons on the TL;DR dataset (Völske et al., 2017); however, both only consider English summarization tasks. Rashkin et al. (2021) focus on attribution, releasing ~4.5K annotations from English summarization, table-to-text, and dialogue datasets; Gekhman et al. (2023) also release attribution annotations for 1.4M summaries, but the labels are machine-generated rather than human-annotated. GENIE (Khashabi et al., 2022) released 17K human evaluations across 5 tasks that includes one English summarization task (XSum).

The only other multilingual summarization evaluation dataset, to the best of our knowledge, is mFACE (Aharoni et al., 2023), which has annotations for 31,500 summaries covering a broader set of languages (45 languages). mFACE focuses on one dataset (XL-Sum) and a smaller set of models than SEAHORSE. In §4 we use mFACE as a comprehensive out-of-domain evaluation set, and view it as complementary to SEAHORSE, which aims to provide large-scale and diverse training data for metrics.

## 6 Conclusion

In this work, we present SEAHORSE, a large-scale multilingual, multifaceted dataset for summarization consisting of 96K human annotations of summaries. Due to its size and scope, SEAHORSE enables the training and evaluation of learnt metrics across several quality dimensions. Our results show that SEAHORSE-trained metrics not only achieve strong performance on our own test set but also generalize to other external and out-of-domain benchmarks: mFACE and TRUE. In the future, we are interested in exploring how SEAHORSE can be used more directly to improve the quality of summarization models and metrics, and hope this paper and the public release of SEAHORSE enables further research on these topics.

## Limitations

The summaries in this work are in 6 languages, and the selection of these languages was based on the number of datasets and articles available for each language. We would like future work to explore the incorporation of low-resource languages, perhaps with the use of crosslingual and fewshot summarization systems. While the raters we worked with in this project went through several rounds of instructions and training, there is a degree of subjectivity inherent in the 6 text quality evaluation tasks and human ratings are noisy, as each individual rater may interpret and rate qualities slightly differently. Finally, the mT5-based metrics presented in this work primarily serve as a demonstration of the potential of the SEAHORSE data for developing summarization metrics; they have not optimized via thorough hyperparameter search, comparing different modeling architectures or approaches, etc. We hope the dataset and experimental results will provide a starting point for this type of exploration in the future.

## Ethics Statement

This work relies on the efforts of human evaluators, who were compensated for their work. The summaries in this work are machine-generated and should not be treated as truth; they may contain misleading or incorrect information. None of the human ratings capture this dimension of the text, as our quality dimensions focus on the relationship between the summary and the source article, not a broader set of information or perspectives. For example, if an article contains a factual error, a summary that contains the same error should be rated as "Yes" for Q4 (attribution) because it is consistent with the article. We used summarization models of varying quality in this work, but all are imperfect and their output should be treated with caution.

## Acknowledgements

We would like to thank Ashwin Kakarla and his team for help with the annotations, as well as Slav Petrov, Hannah Rashkin, and our EMNLP reviewers for their feedback on the paper.

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

# A   Training details

The summarization models were trained on the training split of each summarization dataset, with the exception of palm_1shot, which generated a summary given a single example from the dataset and the input article. The checkpoint for each model was selected using performance on the validation set of each respective dataset, except for t5_base_250 and mt5_small_250, which were only trained for 250 steps. The input length for the T5 and mT5 models was set to 1024, and 2048 for PaLM. The target length was 512.

The SEAHORSE metrics were trained on the SEAHORSE training split, and the best checkpoint was selected based on performance on the validation set. A separate metric was trained for each of the 6 dimensions of quality. We used only "Yes" and "No" ratings for training and testing the SEAHORSE metrics. The input length for the learnt metrics model is 2048. The article and summary are separated with "premise:" and "hypothesis:" tags, respectively, to be consistent with Honovich et al. (2022).

All training and inference was done with the t5x framework (Roberts et al., 2022) and run with TPU accelerators.

# B   Rate of positive responses

Table 9 shows a detailed breakdown of the proportion of responses that were positive (i.e., "Yes"), divided by language, dataset, model, and question. Summaries in languages other than English and produced by smaller models tend to have lower scores, indicating good directions for improving our summarization systems.

While most articles in the dataset were assigned to a subset of the summarization models, some articles were summarized by all 9 summarization systems (or 6 systems for the non-en languages that did not use the T5 models). Specifically in the test set, there were 543 articles that were summarized by all summarization systems. Table 10 shows the positive response rate across those summaries.

## C  SEAHORSE **example summaries and scores**

Figure 4 shows 3 summaries from the SEAHORSE dataset, along with ratings for the attribution (Q4) dimension from the human raters, mt5$_{\text{SEAHORSE}}$, and ROUGE-L.

## D  **Comparison between mT5_large and mT5_xxl**

Table 11 compares the results of two versions of mT5 finetuned on SEAHORSE data, mT5_large and mT5_xxl, on the SEAHORSE and mFACE test sets. Scores are generally close between the two models, but mT5_xxl outperforms the large metric in all cases except one.

**Article:** Take deep, slow breaths through your nose if you feel yourself getting emotional. This will help you calm down and give you something concrete to focus on. [...] Your facial muscles become tense when you cry, and it's natural for you to frown beforehand. Try to relax your frown and release all the tension from your face. You don't have to smile—you're at a funeral, after all—but relaxing your face will help keep you from crying. If you feel your facial muscles tensing up, take a couple deep breaths and relax your shoulders. Relaxing other parts of your body will help you relax your face as well. [...]

**Summary:** Make your face feel emotional. Relax your face. Relax your face.

| **Rater:** 0 | $mt5_{SEAHORSE}$: 0.15 | **ROUGE-L:** 33.59 |
|---|---|---|

**Comments:** While the second (and repeated third) sentence in this summary is supported by the article, the first sentence is not, and both the rater and $mt5_{SEAHORSE}$ rate it accordingly. ROUGE incorrectly rates it highly (in the 80th percentile of its scores).

---

**Article:** UN human rights chief backs Apple in FBI encryption row --- The FBI has ordered the tech giant to assist it with unlocking an iPhone used by San Bernadino gunman Syed Farook. Prince Al Hussein said the law enforcement agency "deserves everyone's full support" in its investigation. However, encryption was essential in the interests of freedom, he added. "There are many ways to investigate whether or not these killers had accomplices besides forcing Apple to create software to undermine the security features of their own phones," he said in a statement. "It is potentially a gift to authoritarian regimes, as well as to criminal hackers. "Encryption and anonymity are needed as enablers of both freedom of expression and opinion, and the right to privacy. Without encryption tools, lives may be endangered." [...]

**Summary:** The UN human rights chief has backed Apple in its row with the FBI over encryption.

| **Rater:** 1 | $mt5_{SEAHORSE}$: 0.98 | **ROUGE-L:** 15.66 |
|---|---|---|

**Comments:** This summary is a rewording of the first line of the article, so it is attributable to the article, which the rater and $mt5_{SEAHORSE}$ agree with. ROUGE rates it low, however (in the 10th percentile).

---

**Article:** Wyoming diplodocus skeleton bought for Denmark museum --- [...] Mystery had surrounded the buyer, but the Denmark museum confirmed on Tuesday it had acquired the skeleton. The museum bought the female dinosaur, nicknamed Misty, for £400,000 ($652,000), following a donation from the Obel Family Foundation. [...] Obel Family Foundation chairman Christen Obel said: "I think it's quite obvious and right that the Natural History Museum of Denmark should own a dinosaur. "So when we suddenly had the opportunity to give the museum this early Christmas present, we jumped at the chance. "Misty is an iconic object that fascinates us, and the dinosaur will certainly create value for the museum for many generations to come."

**Summary:** A diplodocus skeleton has been bought by the Natural History Museum of Denmark.

| **Rater:** 1 | $mt5_{SEAHORSE}$: 0.85 | **ROUGE-L:** 15.82 |
|---|---|---|

**Comments:** Though the first line of the article seems to contradict the summary (*for* the museum vs. *by* the museum), the article later clarifies that it was in fact the museum that bought the dinosaur, so the rater and $mt5_{SEAHORSE}$ are correct. Only the ROUGE metric rates it low (in the 10th percentile).

Figure 4: Example summaries and ratings from the human raters, $mt5_{SEAHORSE}$, and ROUGE-L for attribution (Q4).

## DE "YES" RATE

| Dataset | Model | Q1 | Q2 | Q3 | Q4 | Q5 | Q6 |
|---|---|---|---|---|---|---|---|
| | reference | 0.99 | 0.99 | 0.98 | 0.82 | 0.64 | 0.55 |
| | mt5_small_250 | 0.83 | 0.58 | 0.59 | 0.68 | 0.41 | 0.29 |
| | mt5_small | 0.93 | 0.85 | 0.87 | 0.68 | 0.47 | 0.38 |
| mlsum | mt5_xxl | 0.98 | 0.97 | 0.95 | 0.8 | 0.59 | 0.5 |
| | palm_1shot | 0.93 | 0.93 | 0.9 | 0.83 | 0.73 | 0.66 |
| | palm_finetuned | 0.99 | 0.99 | 0.99 | 0.88 | 0.82 | 0.73 |
| | total | 0.94 | 0.89 | 0.88 | 0.79 | 0.62 | 0.53 |
| | reference | 0.97 | 0.96 | 0.94 | 0.65 | 0.63 | 0.49 |
| | mt5_small_250 | 0.82 | 0.75 | 0.75 | 0.08 | 0.07 | 0.03 |
| | mt5_small | 0.91 | 0.35 | 0.84 | 0.4 | 0.26 | 0.16 |
| wikilingua | mt5_xxl | 0.97 | 0.91 | 0.93 | 0.69 | 0.62 | 0.49 |
| | palm_1shot | 0.76 | 0.72 | 0.73 | 0.63 | 0.53 | 0.42 |
| | palm_finetuned | 0.98 | 0.97 | 0.95 | 0.74 | 0.79 | 0.65 |
| | total | 0.9 | 0.78 | 0.85 | 0.53 | 0.48 | 0.37 |
| total | | 0.92 | 0.84 | 0.87 | 0.66 | 0.55 | 0.45 |

## RU "YES" RATE

| Dataset | Model | Q1 | Q2 | Q3 | Q4 | Q5 | Q6 |
|---|---|---|---|---|---|---|---|
| | reference | 0.99 | 0.98 | 0.94 | 0.48 | 0.82 | 0.44 |
| | mt5_small_250 | 0.4 | 0.21 | 0.29 | 0.2 | 0.25 | 0.1 |
| | mt5_small | 0.73 | 0.58 | 0.57 | 0.27 | 0.47 | 0.19 |
| xlsum | mt5_xxl | 0.95 | 0.93 | 0.83 | 0.44 | 0.76 | 0.4 |
| | palm_1shot | 0.89 | 0.89 | 0.82 | 0.78 | 0.66 | 0.6 |
| | palm_finetuned | 1.0 | 1.0 | 0.98 | 0.68 | 0.83 | 0.6 |
| | total | 0.83 | 0.77 | 0.74 | 0.48 | 0.64 | 0.39 |
| | reference | 0.97 | 0.95 | 0.9 | 0.56 | 0.65 | 0.46 |
| | mt5_small_250 | 0.73 | 0.22 | 0.66 | 0.31 | 0.05 | 0.04 |
| | mt5_small | 0.83 | 0.26 | 0.75 | 0.39 | 0.17 | 0.09 |
| wikilingua | mt5_xxl | 0.96 | 0.92 | 0.85 | 0.54 | 0.61 | 0.45 |
| | palm_1shot | 0.92 | 0.86 | 0.86 | 0.74 | 0.48 | 0.36 |
| | palm_finetuned | 0.93 | 0.93 | 0.89 | 0.66 | 0.59 | 0.51 |
| | total | 0.89 | 0.69 | 0.82 | 0.53 | 0.42 | 0.32 |
| total | | 0.86 | 0.73 | 0.78 | 0.5 | 0.53 | 0.35 |

## EN "YES" RATE

| Dataset | Model | Q1 | Q2 | Q3 | Q4 | Q5 | Q6 |
|---|---|---|---|---|---|---|---|
| | reference | 1.0 | 1.0 | 0.96 | 0.54 | 0.68 | 0.47 |
| | t5_base_250 | 0.96 | 0.88 | 0.89 | 0.32 | 0.43 | 0.24 |
| | t5_base | 0.96 | 0.91 | 0.91 | 0.42 | 0.5 | 0.32 |
| | t5_xxl | 0.99 | 0.98 | 0.97 | 0.58 | 0.64 | 0.47 |
| | mt5_small_250 | 0.7 | 0.47 | 0.57 | 0.17 | 0.2 | 0.09 |
| xsum | mt5_small | 0.84 | 0.68 | 0.75 | 0.17 | 0.24 | 0.12 |
| | mt5_xxl | 0.97 | 0.95 | 0.93 | 0.46 | 0.58 | 0.37 |
| | palm_1shot | 0.97 | 0.96 | 0.91 | 0.48 | 0.55 | 0.39 |
| | palm_finetuned | 0.99 | 0.99 | 0.99 | 0.6 | 0.65 | 0.51 |
| | total | 0.93 | 0.87 | 0.87 | 0.42 | 0.5 | 0.33 |
| | reference | 1.0 | 1.0 | 0.97 | 0.6 | 0.74 | 0.51 |
| | t5_base_250 | 0.98 | 0.93 | 0.92 | 0.59 | 0.59 | 0.43 |
| | t5_base | 0.99 | 0.96 | 0.96 | 0.65 | 0.68 | 0.52 |
| | t5_xxl | 1.0 | 0.99 | 0.97 | 0.68 | 0.72 | 0.54 |
| | mt5_small_250 | 0.74 | 0.53 | 0.59 | 0.29 | 0.24 | 0.15 |
| xlsum | mt5_small | 0.89 | 0.78 | 0.79 | 0.4 | 0.44 | 0.29 |
| | mt5_xxl | 0.99 | 0.98 | 0.94 | 0.62 | 0.73 | 0.52 |
| | palm_1shot | 0.95 | 0.95 | 0.92 | 0.73 | 0.68 | 0.58 |
| | palm_finetuned | 1.0 | 1.0 | 1.0 | 0.62 | 0.69 | 0.45 |
| | total | 0.95 | 0.9 | 0.9 | 0.57 | 0.6 | 0.44 |
| | reference | 0.99 | 0.99 | 0.93 | 0.55 | 0.59 | 0.42 |
| | t5_base_250 | 0.98 | 0.59 | 0.93 | 0.31 | 0.26 | 0.09 |
| | t5_base | 0.98 | 0.89 | 0.93 | 0.67 | 0.57 | 0.45 |
| | t5_xxl | 0.98 | 0.95 | 0.92 | 0.68 | 0.63 | 0.51 |
| | mt5_small_250 | 0.96 | 0.27 | 0.91 | 0.45 | 0.09 | 0.02 |
| wikilingua | mt5_small | 0.95 | 0.65 | 0.88 | 0.52 | 0.37 | 0.19 |
| | mt5_xxl | 1.0 | 0.96 | 0.92 | 0.62 | 0.64 | 0.49 |
| | palm_1shot | 0.98 | 0.95 | 0.94 | 0.8 | 0.58 | 0.49 |
| | palm_finetuned | 0.99 | 0.98 | 0.95 | 0.6 | 0.63 | 0.54 |
| | total | 0.98 | 0.8 | 0.92 | 0.58 | 0.48 | 0.35 |
| total | | 0.95 | 0.86 | 0.9 | 0.53 | 0.53 | 0.38 |

## TR "YES" RATE

| Dataset | Model | Q1 | Q2 | Q3 | Q4 | Q5 | Q6 |
|---|---|---|---|---|---|---|---|
| | reference | 1.0 | 1.0 | 0.88 | 0.46 | 0.82 | 0.43 |
| | mt5_small_250 | 0.59 | 0.41 | 0.34 | 0.23 | 0.33 | 0.17 |
| | mt5_small | 0.85 | 0.72 | 0.57 | 0.35 | 0.49 | 0.29 |
| xlsum | mt5_xxl | 0.99 | 0.98 | 0.83 | 0.54 | 0.78 | 0.49 |
| | palm_1shot | 0.83 | 0.8 | 0.73 | 0.77 | 0.72 | 0.66 |
| | palm_finetuned | 1.0 | 0.99 | 0.9 | 0.62 | 0.83 | 0.57 |
| | total | 0.87 | 0.81 | 0.7 | 0.48 | 0.65 | 0.42 |
| | reference | 0.94 | 0.92 | 0.83 | 0.5 | 0.73 | 0.46 |
| | mt5_small_250 | 0.9 | 0.34 | 0.79 | 0.35 | 0.2 | 0.12 |
| | mt5_small | 0.82 | 0.53 | 0.57 | 0.1 | 0.18 | 0.05 |
| wikilingua | mt5_xxl | 0.93 | 0.89 | 0.77 | 0.44 | 0.61 | 0.35 |
| | palm_1shot | 0.84 | 0.77 | 0.76 | 0.7 | 0.63 | 0.49 |
| | palm_finetuned | 0.94 | 0.93 | 0.87 | 0.69 | 0.74 | 0.62 |
| | total | 0.89 | 0.72 | 0.76 | 0.44 | 0.5 | 0.33 |
| total | | 0.88 | 0.78 | 0.72 | 0.47 | 0.61 | 0.39 |

## VI "YES" RATE

| Dataset | Model | Q1 | Q2 | Q3 | Q4 | Q5 | Q6 |
|---|---|---|---|---|---|---|---|
| | reference | 0.86 | 0.85 | 0.81 | 0.37 | 0.65 | 0.35 |
| | mt5_small_250 | 0.49 | 0.33 | 0.39 | 0.09 | 0.17 | 0.06 |
| | mt5_small | 0.7 | 0.57 | 0.59 | 0.2 | 0.41 | 0.15 |
| xlsum | mt5_xxl | 0.84 | 0.83 | 0.8 | 0.38 | 0.67 | 0.36 |
| | palm_1shot | 0.92 | 0.9 | 0.83 | 0.69 | 0.43 | 0.3 |
| | palm_finetuned | 0.99 | 0.99 | 0.93 | 0.52 | 0.67 | 0.42 |
| | total | 0.8 | 0.75 | 0.73 | 0.37 | 0.51 | 0.28 |
| | reference | 0.98 | 0.97 | 0.94 | 0.57 | 0.71 | 0.51 |
| | mt5_small_250 | 0.82 | 0.28 | 0.78 | 0.25 | 0.1 | 0.06 |
| | mt5_small | 0.91 | 0.28 | 0.87 | 0.31 | 0.25 | 0.13 |
| wikilingua | mt5_xxl | 0.97 | 0.95 | 0.93 | 0.49 | 0.65 | 0.42 |
| | palm_1shot | 0.78 | 0.76 | 0.63 | 0.64 | 0.22 | 0.16 |
| | palm_finetuned | 0.99 | 0.98 | 0.96 | 0.73 | 0.39 | 0.33 |
| | total | 0.91 | 0.7 | 0.86 | 0.49 | 0.39 | 0.27 |
| total | | 0.85 | 0.72 | 0.79 | 0.43 | 0.45 | 0.27 |

## ES "YES" RATE

| Dataset | Model | Q1 | Q2 | Q3 | Q4 | Q5 | Q6 |
|---|---|---|---|---|---|---|---|
| | reference | 0.99 | 0.99 | 0.88 | 0.69 | 0.49 | 0.33 |
| | mt5_small_250 | 0.78 | 0.69 | 0.63 | 0.38 | 0.2 | 0.11 |
| | mt5_small | 0.94 | 0.88 | 0.8 | 0.61 | 0.38 | 0.25 |
| mlsum | mt5_xxl | 0.98 | 0.97 | 0.86 | 0.76 | 0.53 | 0.39 |
| | palm_1shot | 0.73 | 0.72 | 0.27 | 0.41 | 0.45 | 0.32 |
| | palm_finetuned | 0.99 | 0.99 | 0.02 | 0.92 | 0.78 | 0.75 |
| | total | 0.9 | 0.87 | 0.57 | 0.63 | 0.47 | 0.36 |
| | reference | 0.99 | 0.99 | 0.96 | 0.31 | 0.49 | 0.21 |
| | mt5_small_250 | 0.64 | 0.44 | 0.55 | 0.17 | 0.16 | 0.07 |
| | mt5_small | 0.8 | 0.63 | 0.71 | 0.23 | 0.28 | 0.12 |
| xlsum | mt5_xxl | 0.98 | 0.96 | 0.94 | 0.39 | 0.43 | 0.23 |
| | palm_1shot | 0.9 | 0.89 | 0.85 | 0.76 | 0.7 | 0.64 |
| | palm_finetuned | 0.99 | 0.99 | 0.98 | 0.5 | 0.66 | 0.41 |
| | total | 0.88 | 0.81 | 0.83 | 0.39 | 0.45 | 0.28 |
| | reference | 0.99 | 0.97 | 0.96 | 0.5 | 0.62 | 0.35 |
| | mt5_small_250 | 0.75 | 0.61 | 0.73 | 0.16 | 0.08 | 0.03 |
| | mt5_small | 0.95 | 0.37 | 0.92 | 0.42 | 0.28 | 0.11 |
| wikilingua | mt5_xxl | 0.98 | 0.93 | 0.95 | 0.57 | 0.64 | 0.4 |
| | palm_1shot | 0.96 | 0.91 | 0.93 | 0.85 | 0.62 | 0.46 |
| | palm_finetuned | 0.99 | 0.97 | 0.94 | 0.84 | 0.84 | 0.74 |
| | total | 0.93 | 0.79 | 0.9 | 0.55 | 0.51 | 0.34 |
| total | | 0.91 | 0.83 | 0.77 | 0.52 | 0.48 | 0.33 |

Table 9: The percent of "Yes" responses, broken down by language, dataset, model, and question number.

| Model | Q1 | Q2 | Q3 | Q4 | Q5 | Q6 |
|---|---|---|---|---|---|---|
| reference | 0.97 | 0.96 | 0.91 | 0.46 | 0.66 | 0.39 |
| t5_base_250 | 0.96 | 0.9 | 0.9 | 0.44 | 0.48 | 0.31 |
| t5_base | 0.98 | 0.95 | 0.94 | 0.51 | 0.58 | 0.38 |
| t5_xxl | 0.99 | 0.98 | 0.95 | 0.67 | 0.72 | 0.58 |
| mt5_small_250 | 0.64 | 0.45 | 0.5 | 0.26 | 0.24 | 0.12 |
| mt5_small | 0.81 | 0.67 | 0.68 | 0.34 | 0.37 | 0.2 |
| mt5_xxl | 0.95 | 0.94 | 0.89 | 0.5 | 0.66 | 0.37 |
| palm_1shot | 0.93 | 0.88 | 0.86 | 0.75 | 0.56 | 0.44 |
| palm_finetuned | 0.98 | 0.97 | 0.91 | 0.66 | 0.73 | 0.56 |

Table 10: The percent of "Yes" responses for the set of articles that have summaries generated by all systems, broken down by model and question.

| Dataset | Metric | Q1 | | Q2 | | Q3 | | Q4 | | Q5 | | Q6 | |
|---|---|---|---|---|---|---|---|---|---|---|---|---|---|
| | | $\rho$ | roc | $\rho$ | roc | $\rho$ | roc | $\rho$ | roc | $\rho$ | roc | $\rho$ | roc |
| SEAHORSE | mt5_L | 0.44 | 0.88 | 0.74 | 0.97 | 0.37 | 0.81 | 0.55 | 0.82 | 0.46 | 0.78 | 0.45 | 0.77 |
| | mt5_XXL | 0.52 | 0.90 | 0.86 | 0.98 | 0.45 | 0.84 | 0.59 | 0.85 | 0.50 | 0.80 | 0.52 | 0.81 |
| mFACE - | mt5_L | 0.14 | 0.77 | - | - | - | - | 0.48 | 0.78 | - | - | 0.32 | 0.70 |
| 5 langs | mt5_XXL | 0.09 | 0.73 | - | - | - | - | 0.50 | 0.79 | - | - | 0.50 | 0.81 |
| mFACE - | mt5_L | 0.13 | 0.68 | - | - | - | - | 0.46 | 0.77 | - | - | 0.36 | 0.71 |
| all langs | mt5_XXL | 0.15 | 0.70 | - | - | - | - | 0.52 | 0.81 | - | - | 0.40 | 0.74 |

Table 11: Metrics' ability to predict SEAHORSE and mFACE ratings, measured with Pearson's coefficient ($\rho$) and the area under the ROC curve (roc). Q1 maps to "Quality" in the mFACE dataset, Q4 to "Attribution," and Q6 to "Informativeness." mt5_L is a SEAHORSE-finetuned version of mT5_large; mt5_XXL is a SEAHORSE-finetuned version of mT5_xxl.