# OpenReview forum: "SEAHORSE: A Multilingual, Multifaceted Dataset for Summarization Evaluation"
_EMNLP/2023/Conference — EMNLP 2023 Main_

### Official Review · Reviewer_SFUF · 2023-08-03

**Soundness:** 5

**Excitement:**

4: Strong: This paper deepens the understanding of some phenomenon or lowers the barriers to an existing research direction.

**Justification For Ethical Concerns:**

n.a.

**Missing References:**

n.a.

**Paper Topic And Main Contributions:**

The paper "SEAHORSE: A Multilingual, Multifaceted Dataset for Summarization Evaluation" presents a new resource, SEAHORSE. This dataset for multilingual, multifacted summarisation evaluation consists of 96,000 summaries with human ratings that relate to six quality dimensions (comprehensibility, repetition, grammar, attribution, main ideas, conciseness). The resource covers six languages and nine "systems" (one of which is a human-created summary). Four datasets were used to create SEAHORSE (mlsum, xsum, xlsum, wikilingua). SEAHORSE can be used as a benchmark for existing systems or as a dataset to train systems. An evaluation on TRUE and mFACE shows that systems trained with SEAHORSE achieve strong performance. The resource is made freely available (it was also submitted to EMNLP as supplementary material).

**Questions For The Authors:**

A: As the paper talks a lot about text quality, you may want to consider introducing the phrase "text quality" in the title of the paper (just a thought since many researchers working on text quality may not immediately discover your paper as relevant for their own work). If it's too late for that, you may want to stress "text quality" a bit more in the abstract.

**Reasons To Accept:**

- The presented resource fills a gap in summarisation research. To the best of the authors' knowledge (and also to the best of this reviewer's knowledge) it is the largest multilingual and multi-dimensional summarisation evaluation resource.
- The dataset has a non-trivial size, covers multiple languages and human annotations across six different dimensions.
- The paper provides a thorough description of the dataset and its annotation process, it also evaluates the dataset in various regards.
- The paper is very clear and well written.

**Reasons To Reject:**

- This reviewer does not see a single reason to reject this paper.

**Reproducibility:**

5: Could easily reproduce the results.

**Reviewer Confidence:**

4: Quite sure. I tried to check the important points carefully. It's unlikely, though conceivable, that I missed something that should affect my ratings.

**Typos Grammar Style And Presentation Improvements:**

- You use the term "systems" a lot in the abstract and also in the main body of the article. It wasn't immediately clear to me that this also includes human reference summaries. Please consider either changing the word (from "system" to something else) or adding something like "(including human reference summaries and LLM-generated summaries)" to explain this immediately, ideally already in the abstract.
- In Table 1, please consider using the full names of the languages instead of the acronyms.
- In Table 3, please consider adding "comprehensible", "repetition", "grammar" etc. to the Q1, Q2, Q3 etc. column heads.
- Please spell out all numbers between one and ten ("nine" instead of "9", "six" instead of "6" etc.).

---

> ### Author Rebuttal · Authors · 2023-08-28
>
> - Question #1: Thank you for your positive feedback. We have noted your comments about the wording and presentation in parts of the paper (e.g., “systems” and “text quality”), and we will update the paper to reflect these points.

---

### Official Review · Reviewer_8cDd · 2023-08-04

**Soundness:** 4

**Excitement:**

4: Strong: This paper deepens the understanding of some phenomenon or lowers the barriers to an existing research direction.

**Missing References:**

1. You have the question for comprehensibility and filter if that is answered no. I wonder if you considered filtering based on grammar. FEQA: A Question Answering Evaluation Framework for Faithfulness Assessment in Abstractive Summarization removed summaries labeled as ungrammatical to not have them influence the annotation of other dimensions.
2. The dimension of conciseness was a bit confusing for me. Especially when in Table 3, many of the ones that score high for conciseness are very long summaries on average according to Table 2. Maybe it’s a matter of naming, but it seems to capture more of an overall quality (correctly gets information while not being too long)
3. One could argue that attribution should be objective and have higher pairwise accuracy (as discussed in L371) along with grammaticality, whose low agreement is somewhat surprising.
4. Did you try translating the English subset of SEAHORSE into the respectively languages and training the metric on that? To better understand the benefit/necessity of annotated in multiple languages.

**Paper Topic And Main Contributions:**

The paper proposes SEAHORSE, a summarization evaluation dataset that covers 6 languages, 9 systems, and 4 summarization datasets and can be used for metric evaluation and training. Summaries are annotated across 6 dimensions. Metrics are trained for each dimension.

**Reasons To Accept:**

1. The dataset will undoubtedly be useful for the community to benchmark metrics. Its multilingual and multidataset aspects make it a valuable contribution to the community.

**Reasons To Reject:**

No major reasons, but see the questions below.

**Reproducibility:**

3: Could reproduce the results with some difficulty. The settings of parameters are underspecified or subjectively determined; the training/evaluation data are not widely available.

**Reviewer Confidence:**

4: Quite sure. I tried to check the important points carefully. It's unlikely, though conceivable, that I missed something that should affect my ratings.

---

> ### Author Rebuttal · Authors · 2023-08-28
>
> We’re glad you feel the dataset will be useful and a valuable contribution to the research community; thank you for the supportive feedback. Please find answers to each of your questions below:
> - Question #1: Serious grammatical errors that render the text unreadable should also be incomprehensible. But if the annotators were capable of understanding the meaning of the text and the grammar errors were only minor (e.g., a missing comma), we thought it would be better to collect the scores for the other dimensions, as they could always be filtered posthoc by the user.
> - Q2: Yes, conciseness is the dimension that captures if the text is a good summary, not just asking about brevity but whether the main ideas (and *only* the main ideas) are accurately represented in the summary. We will make this definition clearer in the paper and consider an alternative name for the dimension.
> - Q3: Despite our best attempts to clearly define each dimension and its criteria, there was a degree of subjectivity for all dimensions. Attribution seems straightforward, but often there are minor discrepancies between the summary and the article that an annotator might not catch or consider important. For example, one annotator thought that references to a “meltdown” after a software update described in an article were enough to justify the summary mentioning computer system “crash,” while another annotator did not. Grammaticality does have high pairwise agreement (0.89), but a lower Krippendorff’s alpha than Q4-6 because there is a higher likelihood of agreement by chance. As with attribution, ambiguity was a cause of disagreement; annotators had different ideas about what spelling/grammar is correct depending on how they interpreted the summary. For example, for the summary “The Mitchell brothers have been reunited with his father's wife, a soap soap maker has said.” One annotator interpreted “his” as referring to the Mitchell brothers and said it was a grammar mistake and should be plural, while another annotator thought it was referring to the soap maker and was therefore correct.
> - Q4: This is a great idea for follow-up work using Seahorse. We did not look into this as part of this research project, but we agree it would be interesting to see how well translation systems would handle the longer documents and if they would preserve the errors in a way that accurately reflects the ratings from the English text. Thank you for the suggestion!

---

### Official Review · Reviewer_A7j7 · 2023-08-04

**Soundness:** 4

**Excitement:**

4: Strong: This paper deepens the understanding of some phenomenon or lowers the barriers to an existing research direction.

**Paper Topic And Main Contributions:**

The paper introduces a dataset of human ratings on 6 qualitative dimensions for multi-lingual summaries coming from 4 datasets and covering 6 languages generated using 9 different systems. The dataset can help evaluate the learnt metrics that are used to evaluate qualitative aspects of summarization systems and also serve as a resource for training such evaluation metrics.

**Questions For The Authors:**

A. What is the reason behind using binary responses for the questions? Some of the questions can be better described if evaluated on multi-level scale such as Q4, Q5 and Q6. Also, Q6 is highly correlated with Q4 and Q5, why did you decide to keep it?

2. Are the models used to generate schemas finetuned further or are used as is? Please provide more details on different models such as whether/how they are trained, data used for training, etc.

3. Please provide the statistics of the instances you used to generate summaries for such as the number of unique articles you used. What is the percentage of articles from different datasets and what is the distribution of articles from different languages? Do different datasets have similar articles and did you remove the duplicates?

4. The first three questions are almost always positive showing that current LLMs are very good at generating readable, coherent and grammatically correct summaries. Why do you include them in the benchmark and can you think of a way to evaluate these metrics in a more fine-grained fashion? Maybe that can reveal aspects which systems still struggle.

5. Why did you only use one annotator for each question and not multiple annotators and then doing a majority vote? This could further improve the quality of responses.

6. What is the justification for using Pearson’s coefficient and ROC for evaluating the learned metrics?

7. For mFACE informativeness metric, why did you use Q6 as a proxy and not Q5? Based on your description for Q5, that seems to be a better alignment.

**Reasons To Accept:**

1. A large-scale dataset of summaries and human-generated ratings of them on 6 dimensions.
2. In-domain and out-of-domain analysis on the performance of the models trained show the quality of the annotations in the dataset by outperforming the baselines.
3. The dataset can be helpful as it can ease the automatic evaluation of qualitative aspects of generation tasks (text summarization in this case) as well as being a test bed for evaluating current metrics.  The experimental results show that the metrics can outperform baselines both on in-domain and out-of-domain sets.

**Reasons To Reject:**

The models that are used to generate the summaries are not structurally diverse, 6 of the models are from T5 family and the other two are variations of the palm model. Though the number of parameters is different, the types of errors made by these systems might be similar and therefore not many diverse patterns will be learned for predicting each proposed metric.

The six questions about the quality of the generated summaries are only measured on a binary scale. For some of these evaluation questions, a more fine-grained multi-scale scoring can lead to better evaluation of the quality.

It is further shown in the experiments that the first three questions are trivial as LLMs are very good at generating readable, coherent and non-repetitive text. This somehow narrows the contributions of the provided dataset as the annotations for these questions might no longer be useful for future systems.

A further analysis of the error types made by system (failure cases where the wrong label is assigned to an instance) can be insightful.

**Reproducibility:**

3: Could reproduce the results with some difficulty. The settings of parameters are underspecified or subjectively determined; the training/evaluation data are not widely available.

**Reviewer Confidence:**

4: Quite sure. I tried to check the important points carefully. It's unlikely, though conceivable, that I missed something that should affect my ratings.

---

> ### Author Rebuttal · Authors · 2023-08-28
>
> Thank you for your encouraging comments and questions; they have been very helpful for identifying the parts of the paper that require more discussion and motivation. Based on your feedback, we will go back and better motivate the design choices in the dataset, especially around the binary scale and the dimensions of quality. Here are more detailed responses to the issues you raised in the "Reason to reject" section:
> - Reasons to reject #1: We used two families of models to generate the summaries in the dataset: (m)T5 and PaLM. We included versions of these model architectures in different sizes, from different training checkpoints, and with/without finetuning to make sure there is enough diversity of summarization quality in the dataset. We experimented with other model architectures such as LSTM-based seq2seq models, but found the quality too low and inconsistent to be worth annotating. We will add more motivation in the paper about our choice of models.
> - RR2: Multi-scale scoring systems provide more nuance, but they also add complexity and noise. Annotators may interpret the scale differently, and defining every point on a Likert scale for each dimension is not a trivial task. E.g., what should be considered a “3” vs. a “4” grammatical error on a 5-point Likert scale? Given our goal of using this dataset as training data for new metrics and to simplify the annotation task, we decided to treat each dimension as a flag, so if a problem is present in a summary, the annotator gives it a negative score.
> - RR3: While we agree in principle that Q1-3 likely have less future relevance, as demonstrated through their high scores, in many settings these dimensions are not trivial for models. While the largest, finetuned models (m/T5 XXL and PaLM-finetuned models) generally do very well on Q1-3, the smaller and 1-shot models do not. Performance also varies by language, as seen in Table 9; English models receive higher positive rates for Q2&3 than languages like Russian, Turkish, and Vietnamese. LLMs are known to be less fluent in many non-English languages and the multilingual nature of our dataset is a key contribution of our work. We consider it important to have benchmarks for these dimensions of quality for building metrics that are subject to computational or resource constraints, multilingual settings, or where it is important to make sure quality along these dimensions hasn’t degraded, e.g., RL models (Paulus et al., 2018).
> - RR4: To clarify, is the “system” here referring to the trained metrics? We will expand Figure 4 to include more examples of false positives and negatives for the trained metrics; thank you for the suggestion, and please let us know if there are other specific analyses you would like to see – we will incorporate them in the final version of the paper.
>
> Please find answers to the "Questions for the Authors" below:
> - Question #1: See RR2 above about the decision to use binary responses. Q6 is the dimension that captures if the text is actually a good summary, i.e., whether the main ideas (and only the main ideas) from the article are accurately represented in the summary. A negative Q4 or Q5 score should result in a negative Q6 score (as described in the paper and in the annotator instructions), but Q6 can be negative even if both Q4 and Q5 are positive.
> - Q2: We are not sure what “schemas” refers to here, but if it is referring to summaries: yes, all the summarization models (other than PaLM-1shot) are finetuned on the training splits of the respective datasets. More details about the finetuning process are in Appendix A.1; we will move this information into the main paper in the next version and elaborate on it, as space allows.
> - Q3: Thank you for the suggestion! There are 32,367 unique articles in Seahorse. Here is a breakdown of the number of articles by dataset:
> | Language | Dataset | Count |
> |------|:---:|:--:|
> | de | wikilingua | 2999 |
> | de | mlsum | 3359 |
> | en-US | wikilingua | 2383 |
> | en-US | xsum | 894 |
> | en-US | xlsum | 2433 |
> | es-ES | wikilingua | 2183 |
> | es-ES | xlsum | 2231 |
> | es-ES | mlsum | 2235 |
> | ru | wikilingua | 2948 |
> | ru | xlsum | 3298 |
> | tr | wikilingua | 770 |
> | tr | xlsum | 2186 |
> | vi | wikilingua | 1951 |
> | vi | xlsum | 2497 |
>
> All languages consist of summaries from at least 2 datasets: one with news articles (xsum, xlsum, mlsum) and one with how-to articles (wikilingua). These are from two separate sources and are two different tasks; the articles are not similar. We have confirmed there are no duplicates between the news article datasets either.
> - Q4: Please see RR2&3 above.
> - Q5: We used one annotator for each data point to cover a broad, diverse, and large set of summaries given the resources and constraints we had. However, Seahorse also contains a subset of the data with multiple annotations (~9K items, as described in section 3) for measuring dataset quality, which could be used as a high quality subset by using majority voting, as you describe.
> - Q6: These are the key metrics used in the papers we compare with; ROC is used in Honovich et al. (2022) (TRUE benchmark), and Pearson’s coefficient in Aharoni et al. (2023) (mFACE benchmark). Unlike other classification metrics, these metrics are not sensitive to a thresholding value. We will clarify the choice of evaluation metrics, along with appropriate citations, in the next version of the paper.
> - Q7: The definition of “informativeness” in the mFace paper is “Is this summary a good summary of the article?” As discussed in Q1, this is closest to what we capture in Q6: is the generated text actually a good summary? Thank you for pointing this out; we will clarify it in the next version of the paper.
>
> References:
> - Roee Aharoni, Shashi Narayan, Joshua Maynez, Jonathan Herzig, Elizabeth Clark, and Mirella Lapata. Multilingual Summarization with Factual Consistency Evaluation. ACL 2023.
> - Or Honovich, Roee Aharoni, Jonathan Herzig, Hagai Taitelbaum, Doron Kukliansy, Vered Cohen, Thomas Scialom, Idan Szpektor, Avinatan Hassidim, and Yossi Matias. TRUE: Re-evaluating Factual Consistency Evaluation. NAACL 2022.
> - Romain Paulus, Caiming Xiong, and Richard Socher. A deep reinforced model for abstractive summarization. ICLR 2018.

---

### Meta-Review · Area_Chair_8P5S · 2023-09-15

**Recommendation:** 5

**Metareview:**

This paper introduces a new dataset for multilingual, multifaceted summarisation evaluation, consisting of 96,000 summaries with human ratings that relate to six quality dimensions (comprehensibility, repetition, grammar, attribution, main ideas, conciseness). The resource covers six languages and nine "systems" (one of which is a human-created summary).

The reviewers were unanimous in that the proposed dataset will be a very valuable contribution to the community, thanks to its size, multilingual coverage and rich annotation. The paper also appears to be clearly written and easy to follow. The reviewers only identified minor issues with this paper, some of which could be addressed in the rebuttal.

---

### Decision · Program_Chairs · 2023-10-07

**Decision:**

Accept-Main

**Comment:**

This paper introduces a new dataset for multilingual, multifaceted summarisation evaluation, consisting of 96,000 summaries with human ratings that relate to six quality dimensions (comprehensibility, repetition, grammar, attribution, main ideas, conciseness). The resource covers six languages and nine "systems" (one of which is a human-created summary).

The reviewers were unanimous in that the proposed dataset will be a very valuable contribution to the community, thanks to its size, multilingual coverage and rich annotation. The paper also appears to be clearly written and easy to follow. The reviewers only identified minor issues with this paper, some of which could be addressed in the rebuttal.